# First Principle Calculation of Accurate Electronic and Related Properties of Zinc Blende Indium Arsenide (zb-InAs)

**DOI:** 10.3390/ma15103690

**Published:** 2022-05-21

**Authors:** Yacouba Issa Diakite, Yuriy Malozovsky, Cheick Oumar Bamba, Lashounda Franklin, Diola Bagayoko

**Affiliations:** 1Department of Studies and Research (DSR) in Physics, Center of Calculation, Modeling and Simulation (CCMS), College of Sciences and Techniques (CST), University of Sciences, Techniques and Technologies of Bamako (USTTB), Bamako BP 3206, Mali; 2Department of Mathematics and Physics (DMP), Southern University and A&M College, Baton Rouge, LA 70813, USA; 3Department of Electrical and Computer Engineering (DECE), Louisiana State University (LSU), Baton Rouge, LA 70803, USA

**Keywords:** density functional theory, BZW method, indium arsenide, band gap, agreement with experiment

## Abstract

We carried out a density functional theory (DFT) study of the electronic and related properties of zinc blende indium arsenide (zb-InAs). These related properties include the total and partial densities of states and electron and hole effective masses. We utilized the local density approximation (LDA) potential of Ceperley and Alder. Instead of the conventional practice of performing self-consistent calculations with a single basis set, albeit judiciously selected, we do several self-consistent calculations with successively augmented basis sets to search for and reach the ground state of the material. As such, our calculations strictly adhere to the conditions of validity of DFT and the results are fully supported by the theory, which explains the agreement between our findings and corresponding, experimental results. Indeed, unlike some 21 previous ab initio DFT calculations that reported zb-InAs band gaps that are negative or zero, we found the room temperature measured value of 0.360 eV. It is a clear achievement to reproduce not only the locations of the peaks in the valence band density of states, but also the measured values of the electron and hole effective masses. This agreement with experimental results underscores not only the correct description of the band gap, but also of the overall structure of the bands, including their curvatures in the vicinities of the conduction band minimum (CBM) and of the valence band maximum (VBM).

## 1. Introduction

Indium arsenide (InAs) is an extensively studied, direct band gap semiconductor. It is reported to have a high electron mobility, and under normal conditions, it crystallizes in the zinc blende structure. It is a material which has been widely studied due to its important applications in many new technological devices, such as electronic and optoelectronic devices. It is also employed in light emitting diodes, photodetectors, and lasers. The functional characteristics of these electronic and optoelectronic devices depend on the engineering of materials at a practical level, and on a clear understanding of the properties of the material [1]. This importance of zb-InAs partly explains the tens of theoretical studies of its electronic and related properties. It has similarly been studied experimentally by several authors.

Table 1 lists results from previous DFT and other calculations of the electronic properties of zb-InAs. For ab initio local density approximation (LDA) calculations, the first six reported negative band gaps, while the following five found a gap of zero. The next six results obtained with an ab initio or ad hoc LDA potentials range from 0.259 eV to 1.396 eV. Ad hoc potentials lack a predictive capability, as the results depend on the values of the various parameters employed in their construction. The computational approach for these calculations and others in the table is mostly the full potential-linearized augmented plane wave (FP-LAPW). A few authors employed the projected augmented wave (PAW), Gaussian type orbitals (GTO), or the standard pseudopotential and plane wave. These approaches are different implementations of the linear combination of atomic orbitals (LCAO).

For calculations with ab-initio, generalized gradient approximation (GGA) potentials, which are listed after the LDA calculations, the findings for the band gap of zb-InAs are not satisfactory; indeed, out of the first 10 ab initio GGA results for the band gap, the first four are negative and the remaining six are zero. The next 9 results in the table, obtained with ab initio GGA potentials, range from 0.204 eV to 0.56 eV. Eight of these results were obtained with the Engel and Vosko GGA that tends to overestimate band gaps.

The above ab initio calculations are followed in the table by several computations with ad hoc potentials. In particular, three of these calculations with an empirical pseudopotential reported band gaps of 0.499, 0.36 eV, and 0.3637 eV. Even though the empirical pseudopotential method entails fitting, there is still a significant difference between the first of these values and the two others. Nine of the calculations employed a version of the modified Becky and Johnson potential and six used the hybrid potential HSE06 or HSE03. Five calculations with the Green function and Dressed Coulomb Approximation (GWA) were not DFT calculations. Their findings are between 0.31 eV and 0.68 eV, a rather wide range. We reiterate that the calculations in this section that employed ad hoc DFT potentials do not have predictive capabilities.

The theoretical results discussed above and shown in Table 1 disagree among themselves and with the experimental values of the band gap of zb-InAs. Six of these experimental values (shown in the last 10 lines of Table 1) range from 0.417 eV to 0.426 eV, for six low temperature measurements, 0.5 eV for a low temperature finding, and 0.354 eV, 0.356 eV, and 0.360 for three room temperature experiments. The apparent, experimental agreement on a band gap around 0.420 eV for low temperature and around 0.36 eV for room temperature is in stark contrast to case of theoretical studies in which 26 of the reported values of the band gap of zb-InAs are negative or zero. The disagreement among theoretical values from ab initio DFT calculations and between these values and accepted experimental ones is the central motivation for this work. The importance of accurate band gap values for the calculations of other properties of materials underscores this motivation. In particular, dielectric functions, optical properties, densities of states and electron effective masses among others, cannot be correctly calculated with an incorrect band gap.

This study is further motivated by the fact that previous works of our group have accurately described or predicted properties of semiconductors using ab initio DFT potentials [39,40]. This feat was made possible by our use of the Bagayoko, Zhao, and Williams (BZW) method or of its enhancement by Ekuma and Franklin (BZW-EF). Bagayoko has explained that these methods actually search for and reach the ground state of a material without using over-completes basis sets. We do not know of any previous calculations in Table 1 that have employed successive, self-consistent calculations, with increasingly large, augmented basis sets to reach verifiably the ground state of the material. Reaching the ground state is required by the second DFT theorem and is needed for results from a DFT calculation to reflect the full, physical content of the DFT and to agree with experimental results [39].

We have structured this article as follows: After this introduction, focused on the importance of the material and previous theoretical and experimental studies, we describe in Section 2 our computational method along with details pertaining to the replication of our work; Section 3 is devoted to the presentation and discussion of our findings for the band structure and band gap, along with the total and partial densities of states, and electron and hole effective masses; and a conclusion is briefly stated in Section 4.

## 2. Computational Method

In our calculations, the potential used is the local density approximation (LDA) one of Ceperley and Alder [41], as parameterized by Vosko, Wilk and Nusair [42]. We employed the linear combination of atomic orbitals (LCAO). Gaussian functions constitute the radial parts of these orbitals. The software we utilized was developed and refined over a decade at the US Department of Energy’s Ames Laboratory, Ames, Iowa [43]. We utilized an experimental lattice constant for room temperature in our non-relativistic calculations. Our approach is distinguished by the application of the BZW method in the execution of the linear combination of atomic orbitals (LCAO). Our method has the capacity to solve concomitantly and in a self-consistent way two coupled equations. The first is that of Kohn and Sham [44], the second equation is the one generating the charge density in terms of the wave functions of the occupied states. This second equation can be taken as a constraint on the Kohn–Sham equation (KS). Many details on the BZW or even the BZW-EF method can be accessed by consulting previous articles [39,45,46,47,48,49,50,51].

Essentially, the method employs successive, self-consistent calculations, with augmented basis sets to carry out a thorough minimization of the energy functional, as required by the second DFT theorem. If two consecutive calculations produced the same occupied energies, these energies are local minima if the next, consecutive calculation lowers any occupied energies. However, if the next consecutive calculation produces the same occupied energies as the two previous ones, then the occupied energies have reached their absolute minima, i.e., the ground state. The robust and rigorous criterion for ending the calculation is to have three consecutive ones produce identical, occupied energies.

In light of the above summary, our calculations necessarily start with a small basis that accounts for all the electrons in the system under study. This Calculation I is following by Calculation II, with a basis set comprising that of Calculation I plus one orbital representing an excited state. We compare the occupied energies of the two self-consistent calculations; invariably, some occupied energies from Calculation II are lower than their corresponding values from Calculation I. After augmenting the basis set of Calculation II with one orbital, we perform Calculation III. The occupied energies from Calculations II and III are compared. This process continues until three consecutive calculations produce the same occupied energies; that is the rigorous proof of the attainment of the ground state. Two energies are considered to be the same or to be identical if they are equal within the uncertainty of 0.005 eV of our computations. Among these last three consecutive calculations, only the first one, which has the smallest basis set, provides the description of the ground state of the material [39,40,45,46,47,48,49,50,51]. The basis set of this calculation is the optimal basis set. The optimal basis set, upon reaching self-consistency, leads to the ground state charge density of the material. Similarly, basis sets that are larger than the optimal one, and that contain the optimal one, lead to the ground state charge density upon reaching self-consistency. As elaborated upon by our group in our publications referenced above, the use of these larger basis sets also lowers some unoccupied energies; these lowered, unoccupied energies are not due to a physical interaction in the Hamiltonian that did not change from its value obtained with the optimal basis set. Incidentally, given that the occupied energies do not change once we reach the optimal basis set, the extra lowering of some unoccupied energies following the use of larger basis sets containing the optimal one is a plausible explanation of the almost universal underestimations of band gaps and energy gaps by conventional calculations. Indeed, these calculations to date, have employed a single basis set that is deliberately chosen to be large in order to ensure completeness. More often than not, these basis sets are over-complete for the description of the ground state [39] by virtue of the Rayleigh theorem for eigenvalues [52]. Even though over-complete basis sets tend to produce an extra lowering of some unoccupied energies, some can also increase some unoccupied energies. By virtue of the second corollary of the first DFT theorem, the lowered and the increased unoccupied energies no longer belong to the spectrum of the Hamiltonian, a unique functional [39] of the ground state charge density.

Details relevant to the replication of our work follow. InAs is stable in the zinc-blende structure (zb, 3C, space group F$\bar{4}$3m ($T_{d}^{2}$)) [53]. We utilized the measured value of the lattice constant at room temperature, 6.0583 Å. The self-consistent calculations for In^+2^ and As^−2^ provided the orbitals for the solid state calculations. Gaussian functions are in the radial parts of atomic wave functions. We used a set of even-tempered Gaussian exponents, with a minimum of 0.121 and a maximum of 0.81400241 × 10^5^ in atomic units, for In^+2^: 19 Gaussian functions were used for s and p orbitals, and 17 for d orbitals. Likewise, to describe As^−2^, the exponents in the Gaussian functions ranged from 0.2404 to a maximum of 0.349 × 10^5^. The numbers of Gaussian functions for s, p, and d orbitals are the same as those for In^+2^. A mesh of 60 k points, with appropriate weights in the irreducible Brillouin zone, was used in the iterations for self-consistency. The error in the valence charge calculation was −0.0017 for 54 electrons, or −3.15 × 10^−5^ per electron. The criterion for the convergence of the iterations was to have a difference no greater than 10^−5^ between the values of the potentials for two consecutive iterations. The number of iterations for this convergence was around 60. Using the above described methods and related computational details, we studied zb-InAs as discussed in the following section.

## 3. Results and Discussions

We present the successive calculations in Table 2. Columns 1, 2, and 3 of the table show the number of a calculation, the valence state orbitals for In^+2^, and the valence state orbitals for As^−2^, respectively. Columns 4 and 5 show the total number of valence functions in our calculations and the calculated, direct band gap at the Γ point, respectively. The occupied energies from Calculations IV, V and VI are identical within our computational uncertainty of 5 meV. Therefore, Calculation IV provides the DFT description of zb-InAs, as per the above description of our method. In particular, the occupied energies from this calculation are those for the ground state of the material. The resulting charge density is that of the ground state of zb-InAs.

The following three figures show the systematic lowering (minimization) of the occupied energies as the basis set is augmented. This true minimization is not attainable with self-consistent iterations with just one basis set; such calculations lead to a stationary state among an infinite number of such states with no proof that the said state is truly the ground state.

The valence bands in Figure 1 and Figure 2 are unlike the ones in Figure 3. Indeed, the superposition of the valence bands in Figure 3 illustrates that we have reached the absolute minima of the occupied energies, i.e., the ground state of the material. A close examination of the quasi-flat bands below −14 eV points to the bands from Calculation 4 as those representing the ground state as they are lower, almost imperceptibly, than those from Calculation III. Calculation V, and VI do not change the occupied bands from their values from Calculation IV.

Various features of the electronic band structures can be further elucidated using the table of calculated eigenvalues at high symmetry points in the Brillouin zone and the figures for the total and partial densities of states. Figure 4 and Figure 5 represent the total density of state (DOS) and partial densities of states (pDOS) derived from the bands from Calculation IV. Many of parts of our calculated density of state are very close to the corresponding experimental values obtained by X-ray photoemission spectroscopy measurements [54]. Ley et al. [54] reported in their Figure 14, that the peak positions of I_2_, I_1_, P_II_, and P_III_ are −1.7 eV, −2.1 eV, −5.8 eV and −10.5 eV, respectively. From our work, the corresponding values are −1.84 eV, −2.20 eV, −5.20 eV and −10.06 eV, respectively. While one can extract the widths of the various groups of valence bands from the total density of states, they are more readily derived from the content of Table 3 that shows the calculated eigenvalues.

The total valence band width is 14.892 eV. The width of the lowest lying group of bands is 0.344 eV. The widths for the middle and upper most groups of valence bands are 1.867 and 5.294 eV, respectively. These widths, from Table 3 can also be estimated using the content of the figure below for TDOS. Another purpose of Table 3 is to enable comparisons with future experimental measurements using X-ray, ultraviolet (UV) or other spectroscopies.

As per our calculated pDOS in Figure 5, the lowest lying group of valence bands comes almost entirely from In d, with a very small contribution from As p. The middle group of valence bands comes largely from As s with some contributions from In s and In p. The upper most group of valence bands is unquestionably dominated by As p and In s with contributions from In p and d and As s.

We used the electronic band structure from Calculation IV to calculate the effective masses of the electron at the bottom of the conduction band, and of the holes at the top of the valence band. We present our results, along with other theoretical and experimental values, in Table 4. The electron effective masses are indicated by m_e,_ while those for the heavy and light holes are denoted as $m_{hh}\;\mathrm{and}\;m_{lh}$, respectively. The first column shows the effective masses with the specific directions in which they are computed. It is gratifying to note that our calculated effective masses are in a general agreement with experimental values [34,55,56,57] and with some previously calculated ones [6,14]. Of the nine effective masses in Table 4, the only one for which there is a discrepancy between an experimental value and our result is $m_{hh}$(Γ-L), for which we found 0.903 *m*_0_, whereas an experiment reported 0.625 *m*_0_ [35]. However, for this same heavy hole effective mass, another experiment [55] reported 85 *m*_0_, basically in agreement with our finding. The experimentally established [34,55] isotropy of the electron effective mass is reproduced by our calculations; our average value of 0.024 is sandwiched between the experimental values of 0.026 [34] and 0.023 [55].

We should underscore that the above agreement between our calculated effective masses and corresponding experimental values points to the correct rendition, by our calculations, of the curvatures of the bands at the conduction band minimum (CBM) and the valence band maximum (VBM).

A discussion of our results in relation to previous works follows. We first recall that our key motivation for this work has been the resolution of the wide disagreement between well-established, measured band gaps of about 0.420 eV and 0.360 eV for low and room temperatures, respectively, and findings from tens of theoretical studies that uniformly underestimated them. In particular, 26 ab initio LDA or GGA studies reported negative numbers or zero for the band gap of zb-InAs. The section describing our method examines the reasons previous DFT calculations differed from the measured values; none of these calculations carried out a thorough minimization of the energy to reach the ground state in a verifiable manner as required by the second DFT theorem. Consequently, although the output of such a calculation is self-consistent, it is one stationary state out of an infinite number of such states; indeed, there exist infinite number of basis sets that can lead to self-consistent (i.e., stationary) results. Consequently, the chances for a single basis set calculation to produce the ground state energies while avoiding over-complete basis sets are practically zero; the articles by Siti Nur Farhana M. Nasir et al. and H. Ullah et al. are some illustrations [59,60]. In contrast, our calculations explicitly reached the ground state and avoided the use of over-complete basis sets that can spuriously lower unoccupied energies by virtue of the Rayleigh theorem [52].

Our results, due to our strict adherence to the theorems of DFT, reflect the full, physical content of DFT. They agree with available, corresponding experimental data as was the case for the room temperature band gap of 0.360 eV. Our work also obtained the experimentally identified locations of the peaks in the valence density of state; far beyond the band gap, and this agreement signifies an overall correct description of the bands. Further, this overall description of the bands is complemented by our correct rendition curvatures of the bands as per our results for the effective masses that agree with corresponding, experimental ones. As elaborated upon by Bagayoko [39], these agreements with experimental values did not require invoking a derivative discontinuity or a self-interaction correction.

## 4. Conclusions

We carried out self-constituting, ab initio calculations of electronic energy bands, total and partial densities of states, and effective masses for zb-InAs. With the Bagayoko, Zhao, and Williams (BZW) method, we carried out a thorough minimization of the energy to reach the ground state without employing over-complete basis sets. Consequently, our results reflect the full, physical content of the DFT and agree with corresponding, experimental values. It is apparent through this work, and others from our group [39,40,45,46,47,48,49,50,51], that the long-standing failure of DFT calculations to describe or to predict electronic properties of materials is due to the said calculations and not to DFT.

## Figures and Tables

**Figure 1 materials-15-03690-f001:**
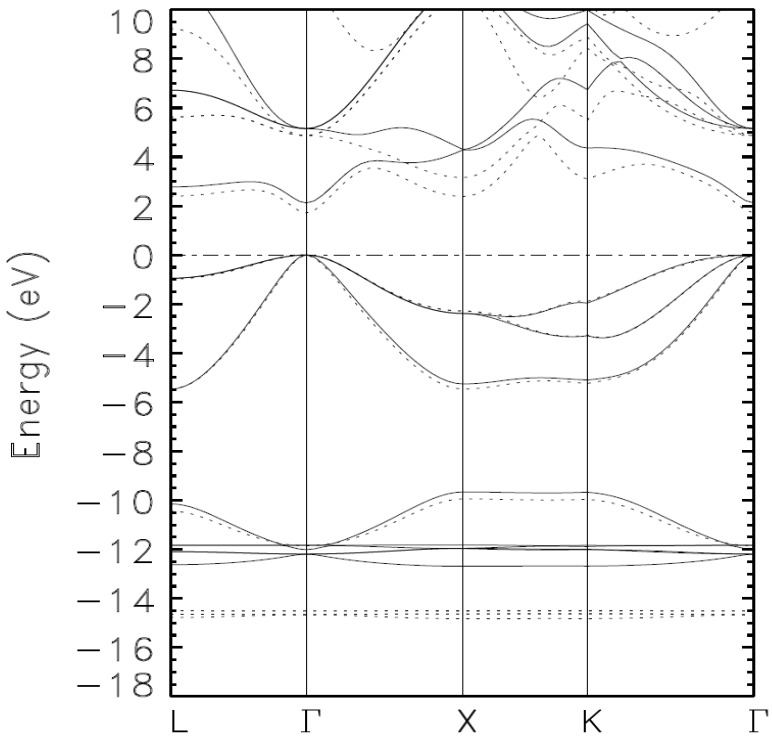
Calculated band structure of zinc blende indium arsenide (zb-InAS) as obtained by the BZW method from calculations I (full line) and II (dashed line). The drastic change in the valence bands, signified by the dashed lines below −14 eV, indicates that the basis set for Calculation I was far from complete for the description of the material.

**Figure 2 materials-15-03690-f002:**
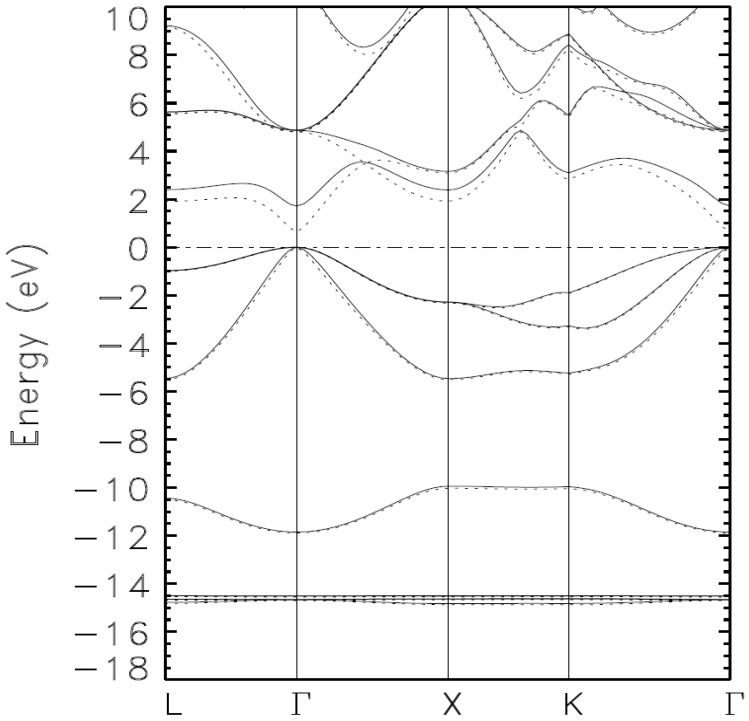
Calculated band structure of zinc blende indium arsenide (zb-InAS) as obtained by the BZW method from calculations II (full line) and III (dashed line). Most of the occupied bands are lowered by Calculation III, albeit only slightly, compared with Calculation II.

**Figure 3 materials-15-03690-f003:**
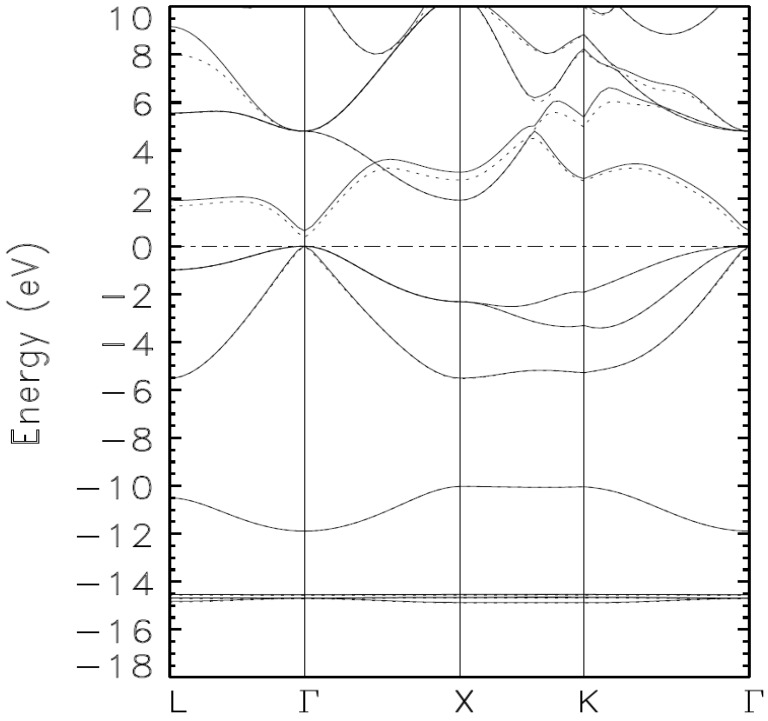
Calculated band structure of zinc blende indium arsenide (zb-InAS) obtained by the BZW method from calculations III (full line) and IV (dashed line).

**Figure 4 materials-15-03690-f004:**
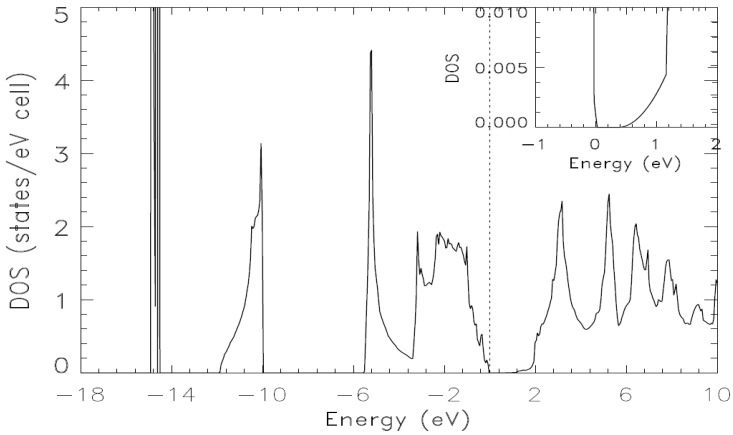
The calculated total density of states of zb-InAs, obtained from the energy bands in from Calculation IV. The zero on the horizontal axis indicates the position of the Fermi level.

**Figure 5 materials-15-03690-f005:**
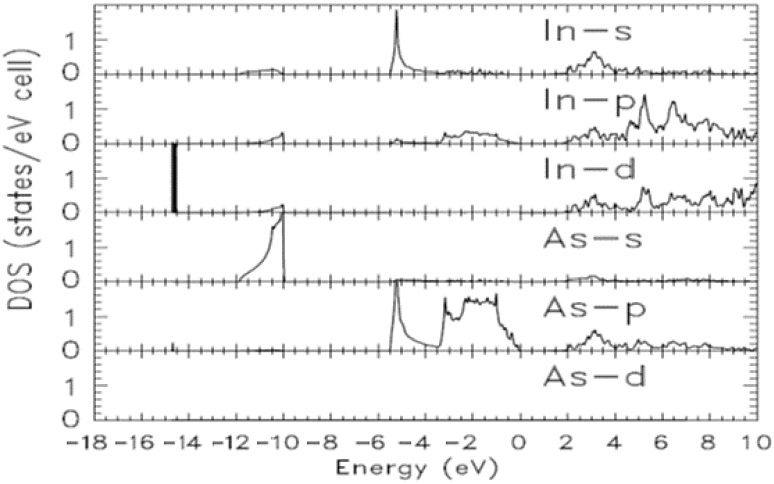
Calculated, partial densities of states (pDOS) for zb-InAs derived from the bands from Calculation IV. The zero on the horizontal axis indicates the position of the Fermi level.

**Table 1 materials-15-03690-t001:** Calculated band gaps (E_g_, in eV) of zinc blende InAs, along with pertinent lattice constants in Angstroms, and experimental values available in the literature. SR = semi-relativistic, R = fully relativistic, CA = Ceperley–Alder, HL = Hedin–Lundqvist.

Computational Formalism	Potentials (DFT and Others)	a (Å)	E_g_ (eV)
FP-LAPW	LDA		−0.64 [a]
FLAPW (R, HL)	LDA	6.058	−0.63 [b]
FLAPW (SR, HL)	LDA	6.058	−0.51 [b]
FLAPW (SR, CA)	LDA	6.058	−0.51 [b]
	LDA		−0.07 [c]
FLAPW (SR, CA)	LDA	6.058	−0.03 [b]
FP-LAPW	LDA	6.0267	0.00 [d]
	LDA	6.0583	0.00 [e]
FP-LMTO	LDA		0.00 [f]
FP-LAPW	LDA	6.030	0.00 [g]
	LSDA		0.00 [h]
	LDA	6.04	0.259 [i]
ab initio pseudo potential	LDA	5.8564	0.4131 [e]
FP-LAPW	mBJ-LDA	6.097	0.47 [j]
FP-LAPW	MBJLDA		0.593 [d]
	MBJLDA		0.61 [k]
	LDA	5.85	1.396 [i]
	GGA		−0.61[l]
	GGA (PBE)		−0.30 [k]
PAW+SOC PBE	GGA	6.194	−0.298 [m]
PAW(PBE)	GGA	6.195	−0.188 [m]
FP-LAPW	GGA-WC	6.0911	0.00 [d]
	WC-GGA		0.00 [n]
FP-LAPW+lo	GGA	6.191	0.00 [o]
FP-LAPW	GGA	6.195	0.00 [g]
FP-LMTO	GGA		0.00 [f]
FP-LAPW	PBE-GGA	6.18922	0.00 [p]
	EV-GGA		0.204 [n]
FP-LAPW	GGA-EV		0.224 [d]
FP-LAPW	GGA	6.194	0.277 [q]
FP-LAPW	GGA-EV with effect spin-orbit		0.31533 [p]
FP-LAPW+lo	EV-GGA		0.34 [o]
FP-LAPW	EV-GGA		0.40 [g]
FP-LAPW	GGA-EV		0.42245 [p]
FP-LAPW	GGA (with SOI)	6.1	0.47 [r]
FP-LAPW	GGA (without SOI)	6.1	0.56 [r]
	mBJ		0.568 [o]
Empirical Pseudopotential Method (EPM)			0.499 [s]
pseudopotential	virtual crystal approximation		0.35 [t]
EPM			0.36 [u]
EPM			0.3637 [v]
PAW	HSE06	6.114	0.544 [m]
PAW	SOC HSE06	6.114	0.420 [m]
PAW	G_0_W_0_ + SOC(HSE06)		0.560 [m]
PAW	G_0_W_0_^TC-TC^ + SOC(HSE06)		0.413 [m]
GTO	SOC HSE03		0.23 [w]
LMTO	scQPGW+SOC		0.68 [x]
	GW		0.31[i]
	GW		0.46 [c]
	HSE		0.39 [h]
	B3LYP		0.55 [h]
	MBJ		0.60 [y]
	TB-MBJ		0.43 [z]
	nTB-MBJ		0.416 [z]
	MBJ		0.57 [r]
	MBJ+PBE		0.46 [r]
	HSE06		0.42 [a’]
FP-LAPW	TB-MBJ with SOC		0.467 [a’]
	TB-MBJ without SOC		0.60 [a’]
	TB-MBJ without SOC		0.615 [b’]
Experiments			
	Low T		0.417 [c’]
			0.426 [d’]
			0.42 [y, e’]
			0.418 [f’]
			0.420 [g’]
	77 K		0.418 [h’]
	Low T		0.5 [i‘]
	At 300 K		0.354 [j’]
			0.356 [h’]
	At 298 K		0.360 ± 0.002 [k’]

[a] Ref [2], [b] Ref [3], [c] Ref [4], [d] Ref [5], [e] Ref [6], [f] Ref [7], [g] Ref [8], [h] Ref [9], [i] Ref [10], [j] Ref [11], [k] Ref [12], [l] Ref [13], [m] Ref [14], [n] Ref [15], [o] Ref [16], [p] Ref [17], [q] Ref [18], [r] Ref [19], [s] Ref [20], [t] Ref [21], [u] Ref [22], [v] Ref [23], [w] Ref [24], [x] Ref [25], [y] Ref [26], [z] Ref [27], [a’] Ref [28], [b’] Ref [29], [c’] Ref [30], [d’] Ref [31], [e’] Ref [32], [f’] Ref [33], [g’] Ref [34], [h’] Ref [35], [i’] Ref [36], [j’] Ref [37], [k’] Ref [38].

**Table 2 materials-15-03690-t002:** The successive, self-consistent calculations of the BZW method for zb-InAs. The calculation corresponding to the optimal basis set is in bold.

Calculation Number	Trial Function for Valence States of In^2+^	Trial Function for Valence States of As^2−^	No. of Functions	Band Gap at Γ (in eV)
Calc I	3d^10^4s^2^4p^6^4d^10^5s^2^5p^0^	3s^2^3p^6^3d^10^4s^2^4p^4^	62	2.123
Calc II	3d^10^4s^2^4p^6^4d^10^5s^2^5p^0^5d^0^	3s^2^3p^6^3d^10^4s^2^4p^4^	72	1.715
Calc III	3d^10^4s^2^4p^6^4d^10^5s^2^5p^0^5d^0^	3s^2^3p^6^3d^10^4s^2^4p^4^5s^0^	74	0.642
**Calc IV**	**3d^10^4s^2^4p^6^4d^10^5s^2^5p^0^5d^0^6s^0^**	**3s^2^3p^6^3d^10^4s^2^4p^4^5s^0^**	**76**	**0.360**
Calc V	3d^10^4s^2^4p^6^4d^10^5s^2^5p^0^ 5d^0^6s^0^	3s^2^3p^6^3d^10^4s^2^4p^4^5s^0^4d^0^	86	0.502
Calc VI	3d^10^4s^2^4p^6^4d^10^5s^2^5p^0^ 5d^0^6s^0^6p^0^	3s^2^3p^6^3d^10^4s^2^4p^4^5s^0^4d^0^	92	0.491

**Table 3 materials-15-03690-t003:** Calculated, electronic energies of zb-InAs at high symmetry points in the Brillouin Zone obtained with the optimal basis set of Calculation IV. We used the experimental lattice constant of 6.0583 Å. This table is partly to enable comparisons with future room temperature, experimental and theoretical results.

L-Point	Γ-Point	X-Point	K-Point
8.016	4.798	10.435	8.840
5.549	4.798	10.435	8.149
5.549	4.798	2.761	4.976
1.679	0.360	1.915	2.731
−0.991	0	−2.321	−1.925
−0.991	0	−2.321	−3.315
−5.523	0	−5.5283	−5.294
−10.501	−11.890	−10.023	−10.036
−14.552	−14.559	−14.548	−14.548
−14.552	−14.559	−14.567	−14.559
−14.694	−14.708	−14.680	−14.669
−14.694	−14.708	−14.680	−14.678
−14.848	−14.708	−14.884	−14.892

**Table 4 materials-15-03690-t004:** Calculated effective masses for zb-InAs (in units of the free electron-mass, $m_{0}$): $\;m_{e}$ indicates an electron effective mass at the bottom of the conduction band; $m_{hh}$, and $m_{lh}$ represent the heavy and light hole effective masses, respectively. Theo: theory, expt: experiment.

	Our Work	Theo [a]	Theo [b]	Theo [c]	$\mathrm{Theo}\;a_{theo}\;[d]$	$\mathrm{Theo}\;a_{exp}\;[d]$	Expt [e]	Expt [f]	Expt [g]	Expt [h]
$m_{e}$ (Γ-L)	0.024	0.027	0.112		0.028	0.018	0.026	0.023 average		
$m_{e}$ (Γ-X)	0.024	0.027	0.094	0.0221	0.028	0.015	0.026			0.023
$m_{e}$ (Γ-K)	0.024	0.027			0.028	0.017	0.026			
$m_{hh}$ (Γ-L)	0.903	0.836			0.878	1.048	0.625	0.85		
$m_{hh}$ (Γ-X)	0.402	0.343	0.353	0.4344	0.381	0.461	0.333	0.35	0.41	0.35
$m_{hh}$ (Γ-L)	0.542	0.623			2.524	2.885	0.513			
$m_{lh}$ (Γ-L)	0.024	0.031			0.026	0.017	0.037			
$m_{lh}$ (Γ-X)	0.023	0.033	0.046	0.0283	0.028	0.015	0.027	0.026	0.026	0.026
$m_{lh}$ (Γ-K)	0.023	0.032			0.026	0.016	0.026			

For Reference [d], a_theo_ and a_exp_ indicate results obtained with a theoretical and an experimental lattice constant. [a] Ref [14], [b] Ref [58], [c] Ref [18], [d] Ref [6], [e] Ref [34], [f] Ref [55], [g] Ref [56], [h] Ref [57].

## Data Availability

Not applicable.

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
