# Peer review of "First Principle Calculation of Accurate Electronic and Related Properties of Zinc Blende Indium Arsenide (zb-InAs)"

_materials, 2022, doi:10.3390/ma15103690_

Round 1
Reviewer 1 Report
The manuscript reports on first principle calculation of the band structure of InAs. Authors follow the approach, which they used successfully at the calculation of some other compounds. The reported results show very good agreement with the experiment. Especially the accord of the band gap is amazing. However, reading the manuscript in detail, I identified multiple weak points, which should be clarified before the manuscript might be accepted for the publication in Materials. Since the objections seem to me important, I request major revision. Specific objections follow.
- Though the title of the manuscript announces the calculation of Transport Properties, there are no real results on this topic. Presented effective masses of free carriers are still details of the band structure, which are important but insufficient for the determination of any transport coefficient. Other important characteristics like for example phonon frequencies or dielectric permittivity are not reported. The title should be amended appropriately giving right information on the paper content to the reader.
- The principle of the method used at the calculations is based on continuously expanding basis set pressuming consecutive minimization of the total energy. Such approach is well approved. However, this effort is not documented in the manuscript. Authors consider the Calculation number IV as the right basis choice deriving their conclusion from the experimentally consistent band gap. Such treatment does not testify on the validity of the results. Combining different basis sets and having knowledge on the right experimentally determined band gap, one can easily imagine that the correct value could be fit. It is necessary to show also the total energy relevant to Calculation numbers and demonstrate that the total energy minimizes at the Calc IV. Simultaneously, authors should explain, why the band gap starts to oscillate at Calcs V and VI.
- Authors neglect relativistic effects in their calculations. This approach could be acceptable at the calculation of band structure of materials composed of light elements. In case of InAs composed of heavier elements, such approach is insufficient. I see in Fig. 1 that the spin orbit splitting of the valence band resulted zero, which is consistent with the approach. However, the right value is about 0.3 eV! It is evident that such crude valence band rearrangement must significantly affect the valence band properties especially around the Gamma point. Respective wave functions are surely critically affected by this fault. It seems to me that the error invoked by this simplification cannot be amended easily and the manuscript can be hardly accepted for the publication unless authors complete the relativistic effects to their theory.
- It would be interesting to verify the consistency of the treatment showing the band structure also at the low temperature close to 0K and check, if the band gap expands in agreement with the experiment.
Minor remarks:
Table 2 caption should be positioned at the head of the table, not flowing in the text.
References are erroneously enumerated by two different schemes.
Ref. 51: There is a strange big symbol BaTiO_3.
Reviewer 2 Report
The work authored by Yacouba Issa Diakite et al, entitled as “First Principles Calculation of Accurate Electronic and Transport Properties of Zinc Blende Indium Arsenide (zb-InAs)” can be publishable in Materials with some revision. In this work, they report electronic and transport properties of zinc blende indium arsenide (zb-InAs). They used different methods of DFT and compared the results with the already reported room temperature data. In fact, from scientific point of view, this work has enough novelty and can be considered for publication in materials after minor amendments as noted below.
- The abstract is very concise and reflect the main idea.
- How the bandgap can be negative? Can author explain this?
- Is it possible to add the required time for these different methods? So, then the author can also say, that particular method is very fast and accurate.
- The quality of Figures needs to be improved.
- The quality of the manuscript can be further improved by considering the relevant literature.
(i) “New Insights into Se/BiVO4 Heterostructure for Photoelectrochemical Water Splitting: A Combined Experimental and DFT Study” Journal of Physical Chemistry C, 2017, 121 (11), 6218–6228.
(ii) Structural and Electronic Properties of Oxygen Defective and Se-Doped p-Type BiVO4(001) Thin Film for the Applications of Photocatalysis” Appl. Catal., B: Environ. 2018, (224), 895-903.
- The is problem with some references. The authors should carefully check all references and do corrections.
Author Response
hello

Reviewer 3 Report
Manuscript: Materials-1638187
The manuscript titled First Principle Calculation of Accurate Electronic and Transport Properties of Zinc Blende Indium Arsenide (zb-InAs) by Y.S.Diakite el al present results on the electronic and transport properties of zinc blende Indium Arsenide. I have following observations:
1// I could not figure out the exact motivation for the present manuscript. The authors have already mentioned about many references (please see Table 1). How these results add-on to the existing literature is not clear in the present form.
2// Moreover, a very good correlation between the theoretical and experimental lattice constants is obtained (line 304-306). Is it a coincidence or it was planned to be like that.
3// Please provide a comparison of the present results with the existing data published by other researchers. A table should be added for this which should help in comparison of the data and results/initial parameters considered by other for calculation.
Based on these observations, I recommend MINOR REVISION for the present paper.
Author Response
Hello

Round 2
Reviewer 1 Report
Authors partly improved the manuscript according my recommendation. However, some issues were not addressed satisfactorily and I request minor revision for the completion of the manuscript for the acceptance in Materials. Respective objections follow.
1. Referencing to #2 in the previous report, the principle of the objection refers to unclear reason, why just the Calculation number IV is defined as the final correct result. It was well documented in added figures 1.,2., and 3., that the band structure changes significantly with the expansion of the basis sets I-II-III-IV. However, why the band gap evolves further and increrases to 0.502 eV and 0.491 eV at Calculations V and VI, respectively? Authors state in line 175: 'Calculations IV, V, and VI are identical, within our computational uncertainty of 5 meV'. If this statement were true, how is it possible that the band gap increases by more than 0.1 eV? It is necessary to comment the results consistently addressing presented values.
2. I accept the explanation of authors why they did not include the relativistic effects in their calculations. I am aware of the complexity of such treatment. I would not protest against used treatment. However, when authors claim in the manuscript title and in the text that their calculations are 'accurate', how should reader interpret this exclamation when important effects were omitted? It is necessary that authors frankly assess the precision of their calculations and discuss the limitation of their approach reporting on its perspective improvement. Arguments in the Cover letter that the used computational method resulted in the correct band gap does not imply any conclusion for unlike materials. There could be simply a coincidence of two different errors, which cancel mutually in a specific material, but could act differently in another one. The word 'accurate' should be expunged in the title. In addition, it is not true that relativistic effects were not considered in recent works. Authors may check for example [P. Romaniello and P. de Boeij, J. Chem. Phys. 127, 174111 (2007)] and [E.S.Kadantsev, https://arxiv.org/pdf/1005.0615.pdf] for this approach.
Author Response
Hello
